# Leak Localization on Cylinder Tank Bottom Using Acoustic Emission

**DOI:** 10.3390/s23010027

**Published:** 2022-12-20

**Authors:** Tuan-Khai Nguyen, Zahoor Ahmad, Jong-Myon Kim

**Affiliations:** 1Department of Electrical, Electronics, and Computer Engineering, University of Ulsan, Ulsan 44610, Republic of Korea; 2PD Technology Cooperation, Ulsan 44610, Republic of Korea

**Keywords:** acoustic emission, constant false alarm rate, cylinder tank bottom, pressure vessels, source localization, time difference of arrival, Voronoi diagram

## Abstract

In this study, a scheme for leak localization on a cylinder tank bottom using acoustic emission (AE) is proposed. This approach provides a means of early failure detection, thus reducing financial damage and hazards to the environment and users. The scheme starts with the hit detection process using a constant false alarm rate (CFAR) and a fixed thresholding method for a time of arrival (TOA) and an end-time determination. The detected hits are then investigated to group those originating from the same AE source together by enforcing an event definition and a similarity score. Afterwards, these newly grouped hits are processed by a time difference of arrival (TDOA) to find the locations of the events. Since the locations of the events alone do not pinpoint the leak location, a data density analysis using a Voronoi diagram is employed to find the area with the highest possibility of a leak’s existence. The proposed method was validated using the Hsu-Nielsen test on a cylinder tank bottom under a one-failed-sensor scenario, which returned a highly accurate result across multiple test locations.

## 1. Introduction

A cylinder tank is a pressure vessel that is used for liquid, vapor, or gas containment in both industrial and civil settings. In general, pressure vessels are known to provide a long, useful lifetime, even with dangerous substances (e.g., acids), if failures are detected and treated early. However, untreated failures can unleash the contained substance, which does not just cause financial damage, but more importantly, precipitate injuries or even fatalities, and turn the surrounding environment hazardous. An example of how appropriate diagnosis and maintenance can greatly extend service life is shown in a hydrofluoric acid sphere tank test [1], which was monitored and fixed properly. In the end, it offered a safe service life of 20 years under a continuous corrosive attack. Afterwards, the same pressure vessel was decommissioned but could still serve in its new role as a water container for years to come.

Leaking on the flat bottom surface is one of the most common problems in a vertical cylinder tank. While in service, it is often not visible or accessible for manual inspection. Diagnostic methods that require tank drainage for inspection are obsolete and can cause unnecessary financial costs while rendering the vessel out of service during the test. Due to this reason, in-service testing has become more favorable in recent years not just for cylinder tanks, but other structures and systems [1,2,3,4] as well. Through this procedure, leak location(s) can be detected early, and maintenance can be performed to prevent further failures and avoid possible environmental contamination.

Acoustic emission (AE) testing offers a non-destructive and in-service means for diagnosis in general structural and machinery health management frameworks [1,2]. When a discontinuity occurs within an object, it emits elastic waves, which are most often in the frequency range of 20 kHz to 1 MHz. This phenomenon is known as an AE, or a low-energy seismic event that can often be observed in nature when a rock fracture occurs. An AE has a number of notable attributes, including non-directionality, in-service testing with little to no downtime, progression tracking and the ability to capture the whole deterioration process with no more than one test. Therefore, it has been widely harnessed for various studies and applications [2,5,6,7,8,9,10,11,12,13,14] across different structures and systems for both industrial and civil use. However, it should be noted that an inspection must be performed before performing an AE test to obtain a priori knowledge of the specimen condition. AE testing does not show existing failures, but rather the occurrence of new ones with a now-or-never attribute. Otherwise, the specimen is assumed to be in the normal working condition at the start of an AE test. In the AE context, the occurrence of a discontinuity that releases elastic waves is referred to as an AE event. When an AE event is recorded by a transducer, it is called an AE hit. Due to the AE being the sole focus of this study, these two terms are henceforth referred to as an event and a hit for convenience.

Source localization has been one of the most important topics regarding structural and machinery health management frameworks. AE testing can allow localization of an active source, given that an ample amount of data is available through collection from the transducer(s). The most notable approaches concerning an AE source location are zone the location technique [1], the signal amplitude difference technique [1] and the timing technique [13,15,16,17,18,19,20]. The zonal approach is one of the more basic techniques for AE localization, which harnesses the idea that the AE source is most likely in the zone of the transducer that returns the highest amplitude, given the assumption of equal transducers’ sensitivity. This technique can offer a simple solution when pinpointing the exact source location is not of utmost importance. However, it is found lacking for the more demanding problems. In the case where structural characteristics are known, the signal amplitude difference of the closest transducers (to the source, determined by the highest amplitude with the same assumption of zonal localization) can be measured and then compared with the known attenuation characteristics. Although this method can give a more detailed and accurate answer than zonal localization, obtaining the information for characteristics can be a huge challenge, especially when a structure or machinery is made of more than one material. The timing technique is one of the more favored approaches in recent years, which uses the difference between the time delays of the same event across separate transducers to derive the source location. Methods that follow this approach can obtain results with high accuracy. However, it is necessary to have precise time difference calculations. The time difference can be estimated through different means, including cross-correlation (a time difference measurement based on the cross-correlation of one discrete or continuous wave in accordance with another), grid search (a time difference measurement achieved by searching the grid zone with the least residual between the calculated and the real distance, either spatially or in time), hit detection and event grouping, etc. In the case of hit detection and event grouping, it can be troublesome because AE events usually happen in bursts with multiple hits happening in a short interval. Cross-correlation-related studies such as [21] adopted a wavelet analysis to extract the useful AE data from noise, then applied a cross-correlation method regarding the geometric positioning principle or investigated multiple weighting function options for the generalized cross-correlation algorithm to estimate the delay time, as in [22]. Other studies following grid search, such as [23], performed a deep analysis of the AE source localization for concrete structures or found the location by leveraging the AE waveform’s reflection, reverberation patterns and their dispersive, multimodal characteristics using only one sensor [24]. In addition, the research in [25] investigated the continuous wavelet transform and the fundamental Lamb wave’s dispersion curves for localization. For the studies regarding the hit detection and the event grouping approach, the burst phenomenon in the AE signal can be analyzed along with the physical wave-propagation model to achieve a promising result [13]. Since our study pursues this approach with a time difference of arrival (TDOA) scheme, a constant false alarm rate (CFAR)-based hit detection and an event grouping method are proposed to solve this problem.

In a real-life leaking situation, AE events are expected to happen not just at the location of the leak, but also in nearby regions due to the turbulent flow of the stored substance. Therefore, the event location alone does not pinpoint the location at which the leak is located, but it can be found through a density analysis of these points. Assuming that the probability density is identical at every event location due to the lack of statistical information regarding the leak, it can be determined that the leak is most likely to be in the spot where the events are densely located [26]. Thus, a Voronoi diagram [27] is employed in this study to search for this region.

In summary, this study proposes the following contributions:A leak localization scheme using AE data, which has not been under investigation to the best of the authors’ knowledge, is proposed with a novel event grouping approach, which gathers hits originating from the same event through similarity measurement.The locations of events are further analyzed using a Voronoi diagram to find the area in which the leak is most likely happening.The study is validated through a case study of a one-failed-sensor scenario.

The following parts of this paper are organized as follows: The methodology is presented in Section 2, whereas Section 3 displays the case study to which the proposed method was applied, and Section 4 provides the conclusion along with future research possibilities.

## 2. Methodology

Prior to a detailed discussion of the methodology, an overview of the proposed process is given in Figure 1. Initially, raw AE data is processed with a CFAR for hit detection, whose results are further investigated with fixed thresholding to determine the TOA and end time. Afterward, hits from the same origin are grouped together using the event definition and similarity score. Using the newly grouped hits, the locations of events can be calculated using TDOA, which are eventually analyzed with a Voronoi diagram to return the final estimated leak location.

### 2.1. AE Hit Detection

The CFAR was originally developed in radar systems for target detection [28]. The principle of a CFAR is to set a power threshold, which distinguishes possible real target hits from the rest (i.e., those considered to originate from spurious sources). The calculation of this threshold is governed by a constant false alarm rate (hence the name) as the trade-off metric between true targets and false ones. In real-life problems, due to many factors, the noise can affect the data both temporally and spatially. Therefore, such difficulties render fixed-threshold-based methods ineffective. A CFAR approaches this problem by adaptively adjusting the threshold level regarding the probability of false alarms, thus lowering the susceptibility to real-life noise. Since its first introduction to radar systems, the CFAR has also been harnessed in other fields due to its advantage in the presence of colored noise with unknown variance. For this research, the average CFAR is employed to detect AE hits from the recorded AE sequences. In order to manage one-dimensional time series such as those investigated in this study, the model displayed in Figure 2 can be used.

The algorithm investigates a cell under test (CUT) through its neighboring cells, which are grouped into guarding and training cells on both sides. The CUT is considered to contain a hit upon exceeding the threshold, which can be derived from the noise power as follows:(1)$$P_{\mathrm{threshold}}=\propto P_{\mathrm{noise}}$$

Given N training cells with $x_{i}$ being the sample in the *i*th training cell and the constant false alarm rate $P_{fa}$, the threshold factor ∝ and the estimated noise power are obtained as follows:(2)$$P_{\mathrm{noise}}=\overset{N}{\underset{i=1}{\sum }}\frac{x_{i}}{N}$$
(3)$$\propto =\frac{N}{\sqrt[{N}]{P_{fa}}}-N$$

As previously discussed, $P_{fa}$ governs the trade-off between the real and false targets. A lower $P_{fa}$ value allows more real targets to be detected at the expense of increased false targets. A higher $P_{fa}$ eliminates a large number of false targets, but real targets might also be ignored. To ensure that the detected targets are not contaminated by false choices, a value of 1 × 10^−4^ was found to provide the best performance. The data is processed in one-second segments without overlapping, and each divided into a collection of 2000 cells. Each CUT is investigated with 10 guarding cells, and 20 training cells and it is considered to contain a hit if the threshold is surpassed.

Since the localization of events is achieved through TDOA, CFAR can only show whether or not a hit is present in the CUT. The next step is to detect the time of arrival (TOA) of hits. In this study, the duration of a hit, which is marked by the TOA and the end time, is determined with a fixed threshold method [23,29,30], which is popular among existing acquisition systems. An appropriate threshold selection is necessary because a low threshold can trigger a premature detection due to noise and a high one can miss the actual TOA by a considerable margin, as can be seen in Figure 3. The first and last threshold crossings are registered as the TOA and end time, respectively. The visualization of this process can be found in Figure 4.

### 2.2. Similarity Score and Event Grouping

Event grouping is often regarded as of lesser importance in comparison to hit detection and event localization in a TDOA-based scheme, even though it is an essential part that directly influences the outcome of localization. In this stage, hits from the same source are grouped together so that later their TOAs can be evaluated with regard to each other to pinpoint the event source. The check for event grouping happens between two hits at a time and consists of two stages: the event definition value (*EDV*) check and the similarity score evaluation.

An *EDV* is a popular fixed-threshold method for event grouping. It sets the maximum time difference between hits that are from the same event. The formula for the *EDV* can be found as follows:(4)$$EDV=\frac{\max\left(D_{\mathrm{sensors}}\right)}{v}$$
with $D_{\mathrm{sensors}}$ consisting of the distances between each pair of sensors and v being the velocity of the elastic wave. Given that in real-life situations events often occur in bursts with multiple discontinuities happening in quick succession, there could be more than one hit existing within the *EDV* range of another. Therefore, it would not be sufficient to employ just the *EDV* for event grouping. Given a reference hit to which others are compared, a list of hits recorded from the next channel and within its *EDV* range are registered for further evaluation.

Afterwards, a similarity score is calculated between the reference hit and each of the hits registered in the list. The authors propose a new similarity score calculation as follows:(5)$$\mathrm{Similarity Score}=\frac{\max\left(\mathrm{cross}-\mathrm{correlation}\left({\mathrm{hit}}_{\mathrm{ref}},\;\mathrm{hit}\right)\right)}{\max\left(\mathrm{auto}-\mathrm{correlation}\left({\mathrm{hit}}_{\mathrm{ref}}\right)\right)}$$

The hit that returns the highest similarity score with the reference one is then considered to be sharing the same source. Subsequently, the search continues across other channels. Any hit registered to one event cannot be registered for another and will be removed from the list indefinitely.

### 2.3. Event Localization Using Time Difference of Arrival

Event localization can be computed for each of the event groups found in the previous subsection. The maximum number of hits that can be registered in one group is equal to the number of sensors deployed in the test. However, the required number of hits for localization on a 2-D plane is three. Therefore, only the three earliest hits from the group are used in this study because the later hits come from the far sensors, which could have been distorted and influenced by an AE noise and other hits.

Given the TOAs $t_{i}$ of the three hits 1–3 and the $\left(x_{i},y_{i}\right)$ coordinates from the sensors which recorded them respectively, the source location $\left(x_{S},y_{S}\right)$ can be found by solving the following set of equations:(6)$$\begin{array}{c} v*\Delta t_{2,1}=\sqrt{{\left(x_{2}-x_{s}\right)}^{2}+{\left(y_{2}-y_{s}\right)}^{2}}-\sqrt{{\left(x_{1}-x_{s}\right)}^{2}+{\left(y_{1}-y_{s}\right)}^{2}}\\ v*\Delta t_{3,1}=\sqrt{{\left(x_{3}-x_{s}\right)}^{2}+{\left(y_{2}-y_{s}\right)}^{2}}-\sqrt{{\left(x_{3}-x_{s}\right)}^{2}+{\left(y_{1}-y_{s}\right)}^{2}} \end{array}$$

The event location results can be further analyzed to find the possible leak source, as presented in the next subsection.

### 2.4. Voronoi Diagram for Data Density Analysis

The Voronoi diagram’s history can be traced back to the 17th century. Throughout the years, it has been known by a few other names, such as Dirichlet tessellation or Thiessen polygons. The core idea of a Voronoi diagram is rather straightforward: given N points in a plane, it tessellates that plane into N convex polygons (also known as regions), each of which is generated from one point (also known as a site), and every point within that region is closer to the generating site than the other ones. The visualization of a Voronoi diagram can be seen in Figure 5.

Each event location is considered a site in a Voronoi diagram and the tessellation is calculated accordingly. Due to the fact that in real-life testing the preceding information concerning the AE sources’ probability density function is unavailable, a uniform distribution is assumed to be present. Therefore, the possible leak position can be indicated by the area with the most densely positioned event locations. Since the density of sites is inversely proportional to the regions’ area [27], the density around a detected event location is calculated as follows:(7)$$D\left(x\right)=A_{x}^{-1}$$
where $A_{x}$ is the area of the region around the event location x that is being considered. By grouping regions that belong to the event locations with the highest density, the possible leak location can be estimated.

## 3. Case Study

### 3.1. Experimental Setup

To verify the proposed method, a Hsu-Nielsen test [31] was employed on a cylinder tank under a one-failed-sensor scenario to imitate a leaking situation. The Hsu-Nielsen test is a test in which a pencil lead of 0.5 mm diameter is broken at a 30° angle against a surface for an AE event generation. The discontinuity upon a lead-breaking occurrence generates an elastic wave which travels along the bottom surface and can be captured by the sensors. For this reason, it is a popular method for an AE event source imitation in multiple experimental setups for different types of machinery or structural failures. In addition, the one-failed-sensor scenario provides a more challenging problem than a conventional setup, in which one of the evenly positioned sensors is considered to be malfunctioning. By doing this, any event that happens in this impaired sensor’s neighborhood, which is supposed to be in the coverage of the failed one, would have to be processed from further channels, thus introducing more attenuation and distortion to the available data. A total of six R6I-AST sensors were attached to the surface of the vertical cylinder tank at a height of 300 mm, and the recordings were taken at a sampling rate of 1 MHz. These sensors from MINTRAS’ offer high sensitivity, and long-driving cable capability with a built-in 40 dB preamplifier and a filter. Moreover, being enclosed in a metal stainless steel housing can provide resistance to electromagnetic/radio frequency interferences and heat stabilization from −35 to 75 Celsius degrees. The “AST” part in the name shows that this genre also supports the integrated Auto Sensor Test, which allows sensor coupling and performance verification even in service.

In this test, the Hsu-Nielsen test was taken multiple times in quick succession at predetermined locations (1–8 and center). Due to the test specimen being in an outdoor setup, other intentional and unintentional AE activities were also randomly introduced during the test. The detailed information concerning the testbed, sensors and location of the Hsu-Nielsen test is illustrated in Figure 6.

Detailed specifications of the specimen are listed below (Table 1):

### 3.2. Result and Discussion

An example of the hit detection process and result are shown in Figure 7. As previously discussed, a hit is detected when the CUT exceeds the threshold calculated from the neighboring cells. However, this only implies that the CUT contains a hit; it does not show the arrival and end time. A fixed threshold of 10% of the peak value, which is calculated in the current CUT, is chosen to mark these important timestamps of the hit. Since there are a large number of hits, Figure 8 shows only the example of a CUT region along with the TOA and end time determination from the respective CUT region. It can be seen that the threshold is capable of detecting the hit’s TOA quite similar to how it would be picked manually.

After the hit detection and TOA determination, the results were then processed for event grouping based on event definition and hit similarity. Subsequently, a TDOA is applied to find the location of the events. As discussed above, the locations of events alone do not sufficiently pinpoint the leak’s location. Therefore, data density analysis with a Voronoi diagram is employed in the final stage in order to localize the area with the highest possibility of leak existence. Figure 9 shows the real Hsu-Nielsen test’s positions previously depicted in Figure 6b: the detected locations of events, and the filtered contours indicating the possible leak region, which are obtained by using the proposed method.

In Figure 9, the likeliness of a leak existence in a region is shown by the color of the contour, with the warmer ones indicating a higher probability and the cooler ones indicating a lower possibility. As it can be seen, the proposed method provides a very close estimation of the Hsu-Nielsen tests’ location across multiple points, even for the ones in the neighborhood of the failed sensor (1, 6, 7, 8). It can also be witnessed that two out of the three real test locations residing outside of the outermost contour are locations 7 and 8 (close to the failed sensor), and even in such cases, the displacement values are insignificant. Some of the abrupt rises in error and large area of possible leak region are due to the introduction of interfering AE activities from the environment and performing Hsu-Nielsen tests.

For a deeper performance analysis, a comparison was executed between the proposed method and a conventional grid search scheme, which is also a popular approach for industrial applications. The localization scheme using a grid search calculates each grid’s residual between the estimated and measured distances to the sensors, then returns the location where this value is minimal. In order to verify the localization accuracy of the proposed method, the displacement is calculated between the test location and: (1) the innermost region (the one covered by the innermost contour with the warmest color, which has the highest probability of leak existence); (2) the outermost region (the entire one covered by the outermost contour with the coolest color, which has a smaller probability of leak existence; if the test location is within this region, then displacement is equal to zero). The results presented in Table 2 show that the proposed method outperforms the conventional grid search localization scheme by a noticeable margin.

## 4. Conclusions

In this paper, the authors presented a leak localization scheme for a cylinder tank bottom with acoustic emission (AE) data. By performing this scheme, leak location can be estimated early, thus allowing the appropriate response to be taken to prevent possible injuries, fatalities and environmental hazards, and minimalize the financial damage. The AE data is initially processed with a constant false alarm rate (CFAR) for hit detection, which is then used to find the hits’ time of arrival (TOA) and end time using the fixed thresholding method. Following this step, hits originating from the same AE source are grouped by applying the event definition and the proposed similarity score. From the obtained results, the sources of AE events were estimated using the time difference of arrival (TDOA). Since AE events can happen in other locations than just the leaking position (due to turbulent flow, etc.), they should be investigated more to derive the estimated leak position. This was analyzed through data density analysis using a Voronoi diagram to obtain the final result.

The proposed scheme was validated in a one-failed-sensor scenario on the cylinder tank bottom with Hsu-Nielsen testing. A total of six sensors were mounted on the tank’s surface, one of which was considered to be malfunctioning. Multiple AE sources were generated on the tank bottom in quick succession at nine different locations, along with random interferences. The obtained result using the proposed method showed a highly precise localization. The localization accuracy is evaluated through the displacement between the real source position and: (1) the innermost region (covered by the innermost contour); (2) the outermost region (the entire one covered by the outermost contour). The first type of displacement returns an average of 0.16 m along with a standard deviation of 0.05 m, and the second returns an average of 0.02 m along with a standard deviation of 0.04 m, which significantly outperforms the conventional grid search localization scheme in a comparison. For future work, the accuracy can be further enhanced by introducing more complicated TOA estimation methods and event localization approaches.

## Figures and Tables

**Figure 1 sensors-23-00027-f001:**
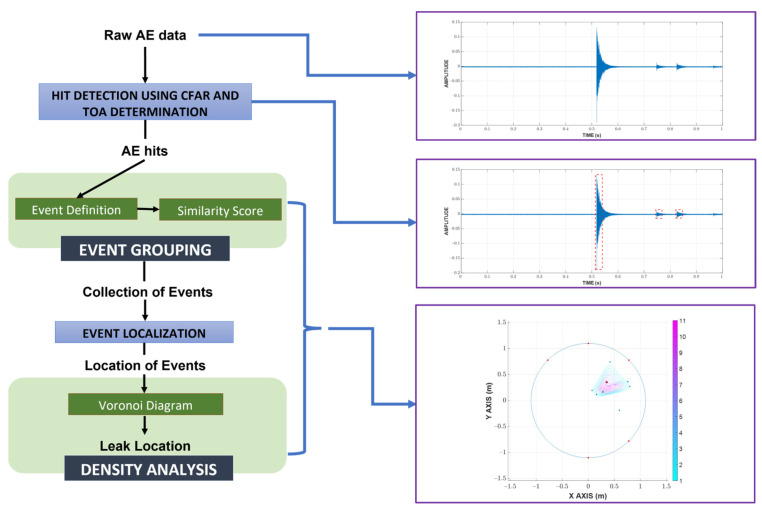
Overview of the proposed localization process.

**Figure 2 sensors-23-00027-f002:**
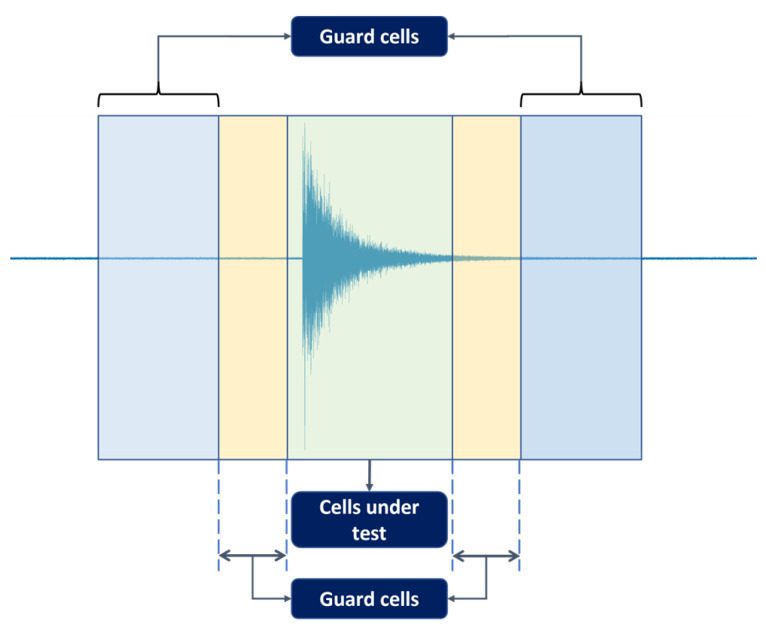
Simplified average CFAR scheme.

**Figure 3 sensors-23-00027-f003:**
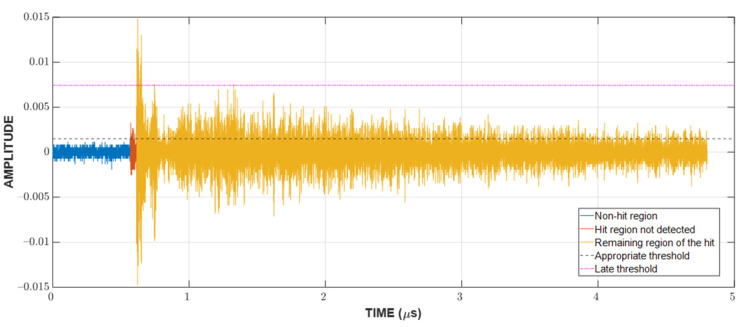
The significance of choosing the appropriate threshold for the TOA determination.

**Figure 4 sensors-23-00027-f004:**
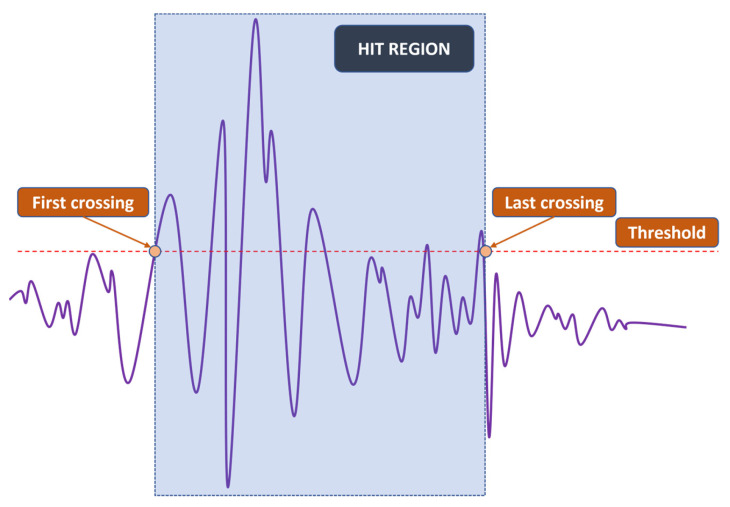
Simplified fix-thresholding TOA and end time determination.

**Figure 5 sensors-23-00027-f005:**
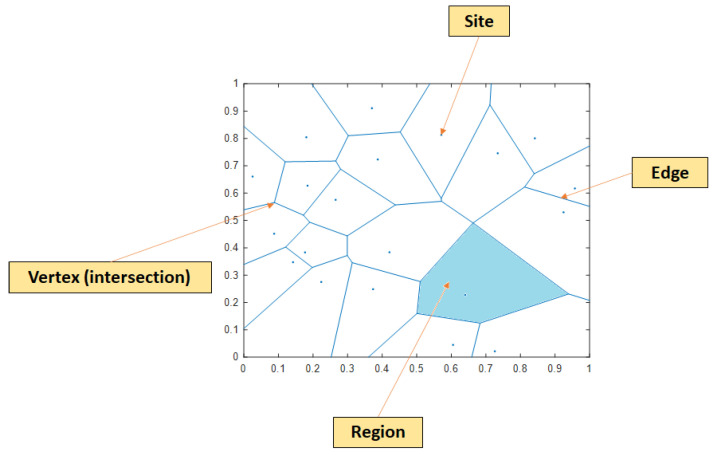
Example of a Voronoi diagram.

**Figure 6 sensors-23-00027-f006:**
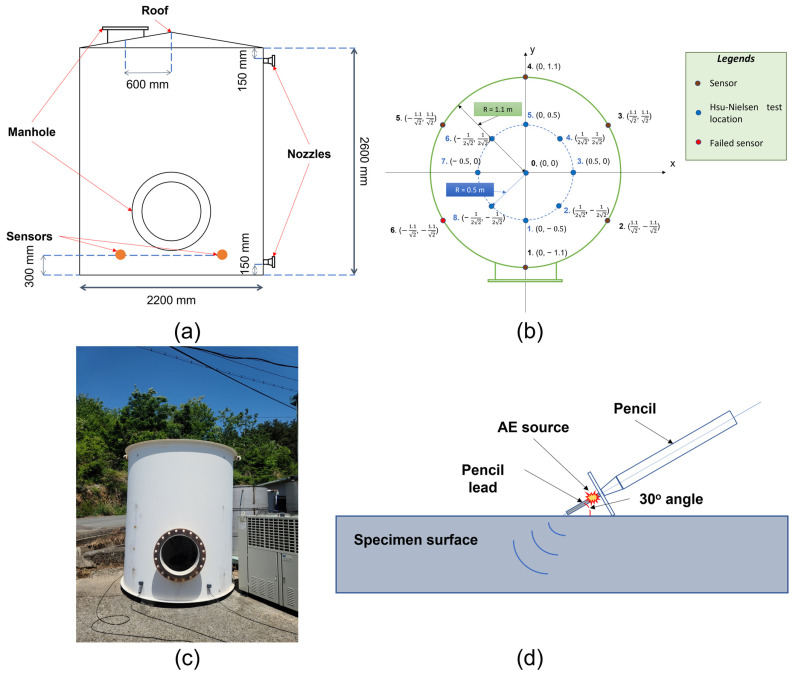
The cylinder tank under test: (**a**) vertical dissection schematic and vertical sensor displacement (**b**) horizontal dissection schematic, sensors and Hsu-Nielsen test location (**c**) pictorial depiction of the test setup (**d**) Hsu-Nielsen test visualization.

**Figure 7 sensors-23-00027-f007:**
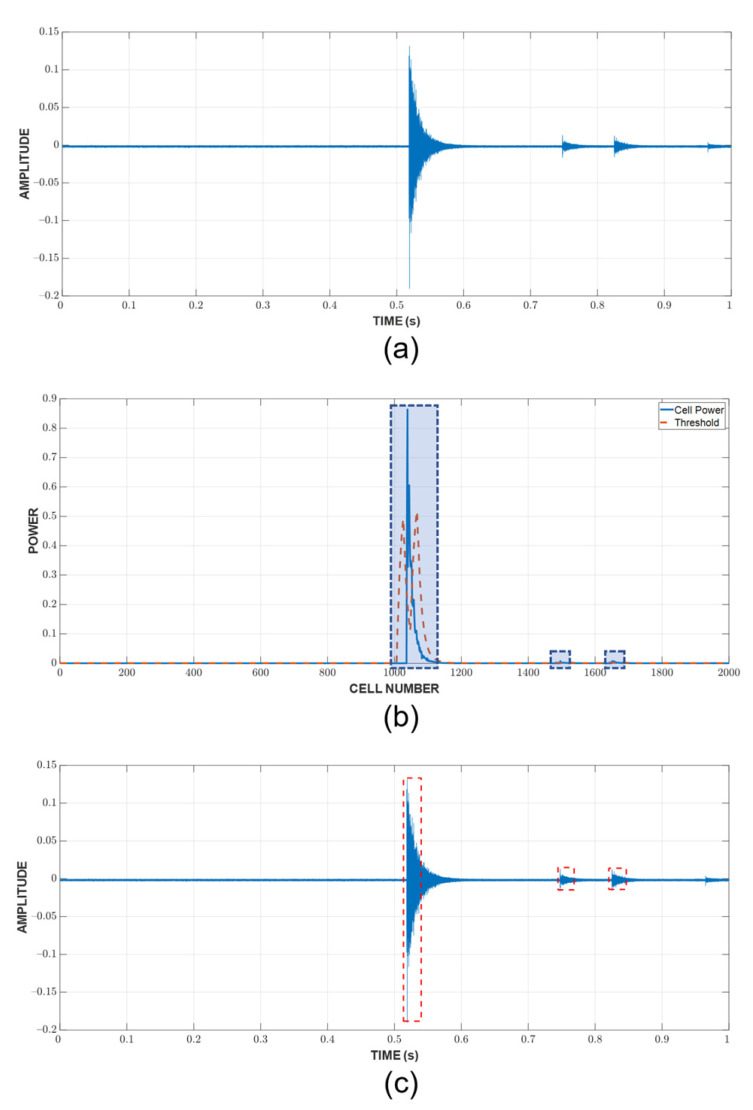
Hit detection process for a random one-second window: (**a**) raw AE data (**b**) cell power versus threshold (**c**) detected hits.

**Figure 8 sensors-23-00027-f008:**
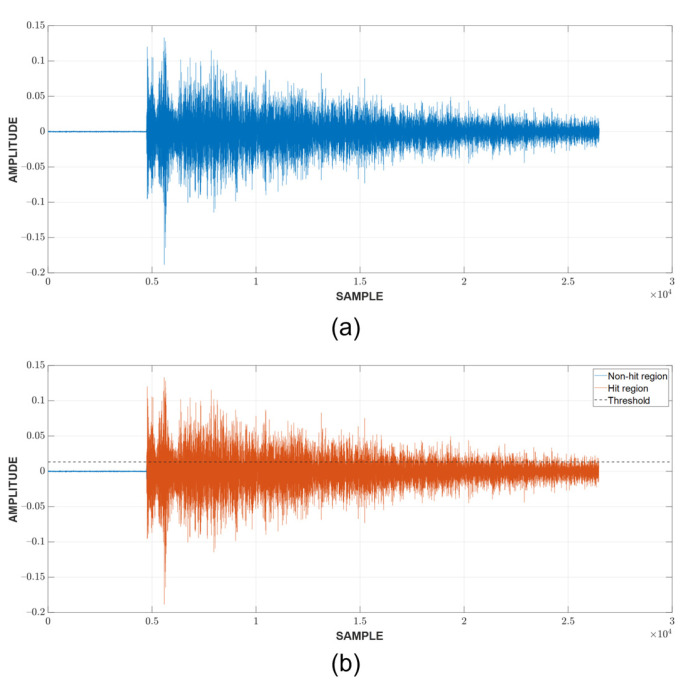
TOA and end time determination: (**a**) CUT containing a hit (**b**) Detected hit.

**Figure 9 sensors-23-00027-f009:**
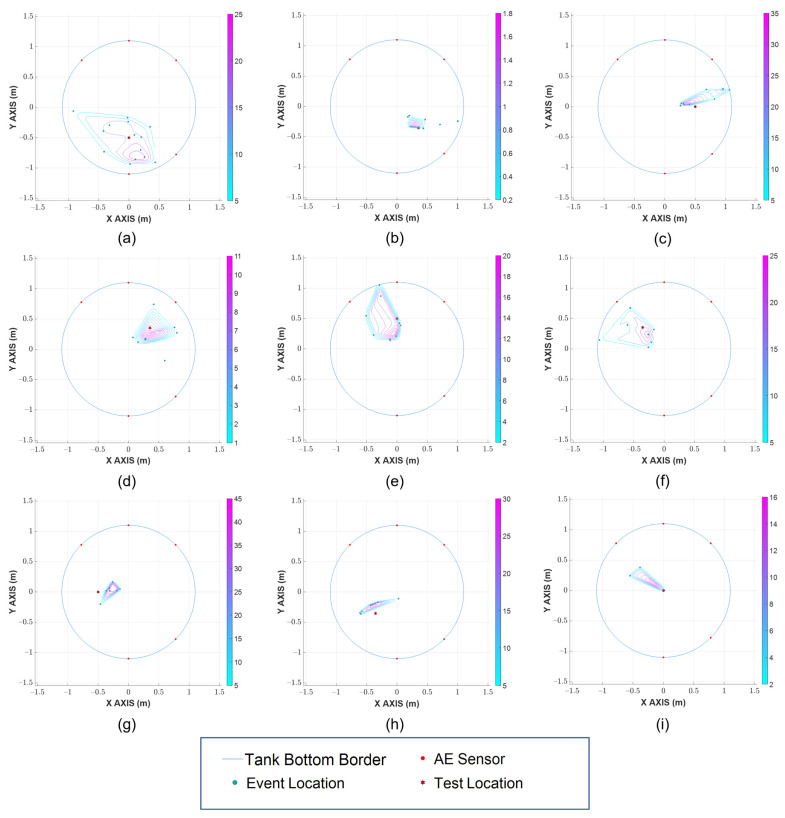
Source localization result in different Hsu-Nielsen test locations: (**a**) 1 (**b**) 2 (**c**) 3 (**d**) 4 (**e**) 5 (**f**) 6 (**g**) 7 (**h**) 8 (**i**) Center.

**Table 1 sensors-23-00027-t001:** Specifications of the cylinder tank.

Parameters/Parts	Details
Dimension (without roof)	2.2 × 2.6 m (diameter × height)
Tank capacity	9.85 m^3^
Tank empty weight	2.1 tons
Tank operating weight	11.95 tons
Shell/roof/bottom material	SA516-70N carbon steel
Flange material	SA105N carbon steel
Nozzle neck material	SA106-B carbon steel
Earthquake design	Yes

**Table 2 sensors-23-00027-t002:** Displacement between the test locations and the results.

Test Location	Conventional Grid Search Localization	The Proposed Method	
		Estimated Displacement to Innermost Region (m)	Estimated Displacement to the Outermost Region (m)
1	0.31	0.24	0
2	0.18	0.11	0
3	0.27	0.17	0.03
4	0.21	0.15	0
5	0.26	0.22	0
6	0.13	0.09	0
7	0.19	0.19	0.12
8	0.22	0.14	0.07
Center	0	0.12	0
	Mean ≈ 0.20/Std ≈ 0.09	Mean ≈ 0.16/Std ≈ 0.05	Mean ≈ 0.02/Std ≈ 0.04

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
