# Peer review of "Leak Localization on Cylinder Tank Bottom Using Acoustic Emission"

_sensors, 2022, doi:10.3390/s23010027_

Round 1

Reviewer 1 Report

1. The conclusion chapter did not demonstrate the conclusion well. The comparation of location detection accuracy with this article and previous research should be given in this article clearly.

2. In the discussion, the fig.8 and fig.9 did not illustrated clearly. This part should be more detailed。

3. the overall arrangement is good and the backgroud and purpose of this article had been clearly explaned.

Reviewer 2 Report

This paper contains extensive study and data regarding leak localization on cylinder tank bottom using acoustic emission. The paper very welly written and have values insight that can be useful for the scientific community and industry. The methodology is scientific, and the paper is organized and written up to journal standard excluding some minor grammatical error in sentence structure. The article not have enough recent literature survey. I suggest minor revision before considering publication

Reviewer 3 Report

The manuscript presents a very future-oriented application of the acoustic emission method for detecting leaks, however there are a few remarks:

1. The Hsu-Nielsen method is not a new method, but rather it is used to indirectly determine the actual value of the acoustic wave velocity for the monitored object, as well as to verify the distance between the source and the sensor. This has been done and confirmed in the article, however, a broader analysis is lacking. Figure 9 lacks a broader description of what these charts actually indicate, why these areas differ. When it comes to a potentially localized source, the amplitude and velocity should also differ. Only one example chart is presented in the article.

Maybe the charts indicated in Figure 9 could be described in more detail?

2. In the drawings there is no description of the axis with the units (Fig.3, Fig.7), as a function of which the amplitude is presented? In duration?

3. The executive summary lacks a brief introduction to the research, which makes it sound like a report. There are also no more detailed conclusions from the research carried out.

4. There are many citations related to the subject in the literature review, however, the article focuses mainly on the general description, maybe it would be worth adding more details.

Round 2

Reviewer 3 Report

I accept the article in this form.